# The Prognostic Value of Plasma Programmed Death Protein-1 (PD-1) and Programmed Death-Ligand 1 (PD-L1) in Patients with Gastrointestinal Stromal Tumor

**DOI:** 10.3390/cancers14235753

**Published:** 2022-11-23

**Authors:** Charlotte Margareta Brinch, Estrid Hogdall, Niels Junker, Holger Jon Moeller, Birgitte Sandfeld-Paulsen, Pieter de Heer, Luit Penninga, Philip Blach Rossen, Anders Krarup-Hansen, Ninna Aggerholm-Pedersen

**Affiliations:** 1Department of Oncology, Copenhagen University Hospital, Herlev and Gentofte Hospital, Borgmester Ib Juuls Vej 1, DK-2730 Herlev, Denmark; 2Department of Pathology, Copenhagen University Hospital, Herlev and Gentofte Hospital, Borgmester Ib Juuls Vej 73, DK-2730 Herlev, Denmark; 3Department of Clinical Medicine, Copenhagen University, DK-2200 Copenhagen, Denmark; 4Department of Clinical Biochemistry, Aarhus University Hospital, Palle Juul-Jensens Boulevard 99, DK-8200 Aarhus, Denmark; 5Department of Clinical Medicine, Aarhus University, DK-8200 Aarhus, Denmark; 6Department of Clinical Biochemistry, Viborg Regional Hospital, Heibergs Allé 4, DK-8800 Viborg, Denmark; 7Department of Surgery and Transplantation, Copenhagen University Hospital, Rigshospitalet, Blegdamsvej 9, DK-2100 Copenhagen, Denmark; 8Department of Oncology, Aarhus University Hospital, Palle Juul-Jensens Boulevard 99, DK-8200 Aarhus, Denmark

**Keywords:** gastrointestinal stromal tumor, PD-1, PD-L1, biomarker

## Abstract

**Simple Summary:**

No soluble biomarker is clinically implemented for patients with Gastrointestinal Stromal Tumor (GIST). High tissue expression of Programmed Death-Ligand 1 (PD-L1) has been associated with a poor prognosis in other cancer types. We aimed to investigate the prognostic value of plasma Programmed Death Protein-1 (PD-1)- and PD-L1 concentrations in patients with GIST. Sensitive immune-assays were used to determine the plasma PD-1 and PD-L1 concentrations in a national study, including 157 patients diagnosed with GIST, independent of disease- and treatment status. Patients with active GIST had significantly higher plasma concentrations of PD-1 and PD-L1 than patients without evidence of disease. Patients with active GIST had the highest plasma concentration of PD-L1 and a significantly poorer prognosis than patients with low concentrations of plasma PD-L1.

**Abstract:**

Background: This study investigates the prognostic value of plasma Programmed Death Protein-1 (PD-1) and Programmed Death-Ligand 1 (PD-L1) concentrations in patients with Gastrointestinal Stromal Tumor (GIST). Methods: Patients with GIST were included (*n* = 157) from the two Danish sarcoma centers, independent of disease- and treatment status. The patients were divided into three subgroups; 1: patients with localized disease who underwent radical surgery; 2: patients with local, locally advanced, or metastatic disease; and 3: patients without measurable disease who had undergone radical surgery. Sensitive electrochemiluminescence immune-assays were used to determine PD-1 and PD-L1 concentration in plasma samples. The primary endpoint was the PFS. Results: No patients progressed in group 1 (*n* = 15), 34 progressed in group 2 (*n* = 122), and three progressed in group 3 (*n* = 20). Significantly higher plasma concentrations of PD-1 (*p* = 0.0023) and PD-L1 (0.012) were found in patients in group 2 compared to PD-1/PD-L1 levels in postoperative plasma samples from patient group 1. Patients with active GIST having a plasma concentration of PD-L1 above the cutoff (225 pg/mL) had a significantly poorer prognosis compared to patients with plasma PD-L1 concentration below the cutoff. Conclusions: Plasma PD-L1 shows potential as a prognostic biomarker in patients with GIST and should be further evaluated.

## 1. Introduction

Gastrointestinal Stromal Tumor (GIST) is a mesenchymal tumor of the gastrointestinal tract, most commonly located in the stomach [1]. The annual incidence is 10–15 per million inhabitants, varying with geographic location [2]. Patients with GIST can remain asymptomatic until the tumor has reached a considerable size, affecting the surroundings unless the tumor causes gastrointestinal bleeding by ulceration [1]. Mutations in either tyrosine-protein kinase KIT (*KIT*) or platelet-derived growth factor receptor A (*PDGFRA*) are characteristic of GIST and are harbored by most GIST cells [3]. The mutation status of *KIT* and *PDGFRA* [4,5], together with the tumor location, tumor size, and mitotic count [6], are prognostic factors and aid in clinical decisions on whether to offer medical oncological treatment to the patients before and after surgery [7,8]. The medical oncological treatment comprises tyrosine kinase inhibitors primarily imatinib, and is used in neoadjuvant, adjuvant, and palliative settings [7,8]. To overcome secondary resistance to TKI treatment, a hypothetic course of action could be targeting the immune system, which is effective in other types of cancer, such as lung cancer [9]. Today, however, immune therapy is not used for treating patients with GIST.

In GIST, two types of infiltrating immune cells are well described: macrophages and T-cells [10]. Programmed death protein-1 (PD-1) expressing T-cells bind to Programmed death-ligand 1 (PD-L1) presented on either a tumor cell or an antigen-presenting cell [11]. The activation of this pathway negatively affects the T-cells, leading to T-cell dysfunction or apoptosis [11], just as tumor cells can avoid undergoing apoptosis when the PD-1/PD-L1 pathway is activated [11]. 

It has been reported that the tumor expression of PD-L1 is associated with a poor prognosis for several types of cancer, such as pancreatic cancer [12], gastric cancer [13], hepatocellular carcinoma [14], esophageal cancer [15], and renal cell carcinoma [16]. However, studies investigating the prognostic value of PD-L1 protein expression using tissue from patients diagnosed with GIST have shown divergent results [17,18,19,20]. 

In a study by Fanale D et al. (2021) [21], patients with untreated, metastatic GIST (*n* = 30) harboring a *KIT* exon 11 aberration were included. In this highly selected cohort, the plasma concentrations of PD-1 and PD-L1 were evaluated. An association was found between a shorter progression-free survival (PFS) and plasma concentrations of PD-1 and PD-L1. This study had a predetermined threshold for PD-1 and PD-L1 plasma concentrations. 

To date, no soluble biomarker is used in clinical diagnostic or prognostic decisions in patients with GIST. Therefore, we aimed to investigate the prognostic value of plasma PD-1 and PD-L1 concentrations in a nationwide, prospectively included cohort of patients diagnosed with GIST independently of disease or treatment status.

## 2. Materials and Methods

This prospective, non-interventional, explorative study aimed to investigate the prognostic value of plasma PD-1 and PD-L1 concentrations in patients with GIST. 

The Regional Ethics Committee (H-18029854) and the Head of the Knowledge Center on Data Protection Compliance (P-2019-706) approved the study. The study was performed according to the latest revised Helsinki declaration and Danish Legislation. All patients included in the study provided written informed consent.

### 2.1. Patients

Patients were included from January 2019 to December 2021 at the Department of Oncology, Herlev, and Gentofte Hospital, the Department of Oncology, Aarhus University Hospital, and the Department of Surgery and Transplantation, Rigshospitalet, in Denmark. The oncological departments included patients with GIST independent of disease- or treatment status. All patients with a GIST of ≥2 cm planned for surgery immediately or after neoadjuvant treatment were included. Primary tumor size was determined on surgery tumor specimens for patients undergoing primary surgery or on a CT scan for patients undergoing surgery after receiving neoadjuvant imatinib. The following patients were excluded: patients ending adjuvant treatment > two years ago and patients who started the adjuvant treatment 0–30 months ago. 

The patients were divided into three groups depending on disease status: (1) patients with local disease who underwent radical surgery, (2) patients with local, locally advanced, microscopic, or macroscopic metastatic disease, and (3) patients without measurable disease (patients radically resected for localized GIST and in adjuvant treatment or patients in surveillance after completed adjuvant treatment) (Figure 1). Blood samples were collected preoperatively and one day postoperatively, and for patients followed at the oncological departments, collected at inclusion and at times for control CT scans, typically every third month. The blood samples collected from patients in group 1 were divided into preoperative (group 1A) and postoperative samples (group 1B).

### 2.2. Samples Handling

Blood samples were collected in 3.5 mL sodium citrate tubes and handled through the Danish CancerBiobank, Bio- and GenomeBank, Denmark. There was a four-hour limit from blood sampling to centrifugation. The centrifugation was performed at 2000× *g* or 2500× *g* for 10 min to isolate plasma. There was a two-hour limit from processing the blood samples to storage. All samples were handled within 6 h during one working day. The plasma was stored at −80 °C until use for the study. The plasma had been thawed once during the storage period. 

### 2.3. Determination of PD-1 and PD-L1 

The PD-1 and PD-L1 analyses were performed using U-PLEX human PD-1 (Epitope 1) and PD-L1 (Epitope 1) assays from Meso Scale Discovery (Rockville, MD, USA) MESO QuickPlex SQ reader using electrochemiluminescence according to the descriptions from the supplier. Initially, 96-well U-PLEX plates were coated with a color-coded linker for each antibody. The linker and biotinylated capture antibody were coupled through mixing and incubation at room temperature. A stop solution was added to the mix to stop the linker-and antibody reaction. The solution was then applied to the plate, shaken for one hour at room temperature, and washed three times. 

The plasma samples were thawed and then centrifuged at 2000× *g* for 3 min. Fifty µL of plasma (diluted 1:4) or calibrator standard was added to each well on the plate, and the plate was incubated at room temperature with shaking for two hours. The plate was washed three times, and subsequently, detection antibodies conjugated with SULFO-TAG were added to each well and incubated at room temperature with shaking for one hour. 

After washing, MSD GOLD Read Buffer B was added to the plate, catalyzing the electro-chemiluminescent reaction. The plate was read using the MESO QuickPlex SQ reader, and emitted light signals were converted to concentrations on the calibration curve using the built-in software. 

Samples with results above the fit curve range (*n* = 2) were excluded from the dataset since the calculated concentrations were extrapolated, and data should be used with caution. 

### 2.4. Data Analysis

Baseline characteristics are presented as numbers with percentages or as medians with 5% and 95% percentile. Categorical variables were compared by the Student’s *t*-test, whereas continuous variables were compared by the Kruskal–Wallis H test. For comparison of plasma concentrations obtained from one patient at different time points (pre- and postoperative as well as at inclusion and at progression), we used the non-parametric Wilcoxon matched pairs signed rank test since the difference in means was not normally distributed. 

The primary endpoint was PFS. Progression of GIST was evaluated by the Response Evaluation Criteria in Solid Tumors 1.1. [22]. Death due to GIST was also interpreted as the progression of GIST. The cutoff date for data analysis was 1 September 2022. 

The Kaplan–Meier method and log-rank test were applied to compare median PFS. Crude and adjusted hazard ratios (HRs) were estimated using the Cox regression model. The multivariate analysis included the plasma PD-1 or PD-L1 concentrations (pg/mL), sex, and age at inclusion. 

The patient groups of interest were divided into low or high concentrations based on the group’s median values of PD-1 or PD-L1. Furthermore, the groups were divided into quartiles based on the plasma concentrations of PD-1 and PD-L1.

Stata v. 17 was used for data analysis, and Graphpad Prism (version 9) was used for visualizing median values and changes in PD-1 and PD-L1 values over time. For all analyses, a significance level of 0.05 was used.

## 3. Results

### 3.1. Patient Characteristics

A total of 157 patients were included in this study, 15 patients in group 1, 122 in group 2, and 20 in group 3 (Figure 1). Table 1 shows patient and disease characteristics at the time of inclusion for patients. There was an equal distribution of men and women (49.7% vs. 50.3%). The median age at inclusion was 69 years (20 to 92 years). There was no statistically significant difference in the age at inclusion between the sexes. In group 1, no patients progressed or died from GIST during the follow-up period, but in groups 2 and 3, thirty-four patients and three patients died, respectively. The median follow-up time was 2.37 years.

### 3.2. Plasma PD-1 and PD-L1

The sex was not significantly associated with plasma PD-1 concentration (plasma PD-1 concentration for males: median 222 pg/mL, P5: 114 pg/mL, P95: 555 pg/mL; plasma PD-1 concentration for females: median 186 pg/mL, P5: 95 pg/mL, P95: 540 pg/mL, *p* = 0.056) but was significantly associated with plasma PD-L1 concentration (plasma PD-L1 concentration for males: median 178.5 pg/mL, P5: 122 pg/mL, P95: 294 pg/mL; plasma PD-L1 concentration for females: median 156 pg/mL, P5: 98.4 pg/mL, P95: 261 pg/mL, *p* = 0.0068), with males having higher plasma concentrations. This was tested using the Kruskal–Wallis test. However, the proportional risks assumption was still fulfilled in the overall model used in the multivariate analysis. Age at inclusion was not associated with the plasma PD-1 or PD-L1 level.

Patients with local, locally advanced, or metastatic GIST (group 2) had significantly higher median concentrations of both plasma PD-1 and plasma PD-L1 than patients that had undergone radical surgery (group 1B) (Table 2 and Figure 2). Furthermore, after stratifying on the disease status, we found a significant difference in the plasma concentration of PD-1 (*p* = 0.033) but not for PD-L1 (*p* = 0.098) (Table 2).

The plasma PD-1 and PD-L1 concentrations were not found to have prognostic value in either the univariate- or multivariate analyses within group 2 patients, Table 3. Sex or age at inclusion alone did not significantly impact the time to progression of GIST (*p* = 0.86 and *p* = 0.14, respectively). Figure 3 shows the Kaplan–Meier plots of PFS for patients in group 2, stratified by the PD-1 and PD-L1 quartile concentrations. Patients in group 2 with the highest plasma concentration of PD-L1 (>225 pg/mL corresponding to patients with a plasma concentration of PD-L1 in the top quartile) had the shortest PFS (HR: 2.13, 95% CI 1.05–4.31, *p* = 0.036). This was confirmed in a multivariate analysis, including age at inclusion and sex (HR: 2.28, 95% CI 1.12–4.65, *p* = 0.023). No such association was found for the plasma concentration of PD-1 (HR: 1.70, 95% CI 0.79–3.64, *p* = 0.17). Since group 2 includes patients with local, locally advanced, micro-, and macro metastatic disease, another multivariate analysis, including disease status, was also performed. The results from the multivariate analysis did not change significantly when including disease status (PD-L1: HR: 2.24, 95% CI 1.09–4.59, *p* = 0.028; PD-1: HR: 1.95, 95% CI 0.90–4.19, *p* = 0.088).

For patients with blood samples available from the time of inclusion and time of progression (*n* = 21), the plasma PD-1 and PD-L1 concentrations from the two-time points were compared (Table 4). The median plasma concentration of PD-L1 was higher at progression than at the time of inclusion, and the difference pointed towards a tendency (*p* = 0.062). No such relation was found regarding the plasma PD-1 concentration. Figure 4 illustrates the plasma concentrations of PD-1 (Figure 4A) and PD-L1 (Figure 4B) over time for the 21 patients having an available blood sample collected at progression. Four patients had a second progression sample, and three had a third progression sample available (Figure 4).

The plasma PD-1 concentration was found to be significantly lower postoperative compared to preoperative, *p* = 0.024 (Table 5 and Figure 5 and Figure 6). No such relation was found for the plasma PD-L1 concentration.

## 4. Discussion

In this study, we evaluated soluble PD-1 and PD-L1 in blood samples from a unique cohort of 157 patients with GIST to explore the prognostic potential of these biomarkers. 

Our study showed that patients with local, locally advanced, or metastatic GIST had significantly higher plasma concentrations of PD-1 and PD-L1 than patients without evidence of disease after radical resection of GIST. The plasma concentration could not distinguish patients with local, locally advanced, micro-, or macro-metastatic disease from each other. Furthermore, we found that PD-L1 plasma concentration could hold a prognostic value in patients with an active GIST, as the patients with the highest plasma concentrations of PD-L1 (>225 pg/mL) had a poorer prognosis compared to patients having a lower plasma PD-L1 concentration (≤225 pg/mL). No such relation was found for plasma PD-1 concentrations. Moreover, a significantly lower plasma concentration of PD-1 postoperative than preoperative was found for patients undergoing radical resection. 

In several types of cancer, a high PD-L1 tumor expression in tissue is associated with a poor prognosis [12,13,14,15,16]. Several studies have investigated the association between PD-L1 expression in GIST tumor specimens and prognosis (Table 6). The results point in different directions regarding the impact of PD-L1 expression on the prognosis. Additionally, the relation between the PD-L1 expression and the number of CD8+ T-cells diverges between the studies. 

One study by Fanale D et al. [21] investigated the prognostic value of the plasma concentrations of PD-1 and PD-L1 in untreated patients with metastatic GIST (*n* = 30), harboring a *KIT* exon 11 aberrations using a customer-developed enzyme-linked immunosorbent assays (ELISAs). Receiver Operating Characteristics curves were used to determine threshold concentrations for each marker to separate patients with a median PFS< and >36 months (short- and long-term survivors). At the best concentrations’ threshold for PD-1 (8.1 ng/mL) and PD-L1 (0.7 ng/mL), the Area Under the Curve was 0.968 (*p* < 0.001) and 1.0 (*p* < 0.001), respectively. The study showed that patients with plasma concentrations of the markers below the predetermined threshold had about 20 months longer median PFS than patients with plasma concentrations above the threshold. In our study, the prognostic value of plasma PD-1 and PD-L1 concentrations were investigated in a clinical setting in a high number of unselected patients with GIST. We found lower plasma concentrations of both PD-1 and PD-L1 compared to Fanale D et al. This may be due to calibration differences between our commercial assay and the in-house assays used in the study by Fanale D et al. [21]. Furthermore, the patients in the study by Fanale et al., had not yet started oncological treatment with imatinib, whereas most patients in our study with active GIST were in oncological treatment, which can affect the results. The two studies also differ in the strategy of choosing cutoff PD-1 and PD-L1 plasma concentrations. Fanale et al. searched for the optimal cutoff, whereas we explored whether higher concentrations had a different prognosis than a lower concentration based on the median and quartile values. 

Since patients in lifelong treatment with TKIs eventually acquire secondary resistance [24], targeting the immune system could be a possible course of action. To our knowledge, three studies of checkpoint inhibitors in patients with GIST have been performed with published and, unfortunately, not convincing results (Table 7). These studies reported a median PFS between 1.5 to 2.9 months [25,26] for patients with GIST and a 6-month non-progression rate of 11% [27]. The median PFS is therefore significantly lower than the median PFS of 4.8 months for third-line treatment with regorafenib in patients with GIST [28].

Several ongoing studies are investigating checkpoint inhibitors alone or in combination with TKIs [30] in patients with GIST. The combination treatment is theoretically a promising approach since patients continue progressing beyond several lines of TKI treatment, and imatinib also has an immunologic effect [31]. Imatinib acted on dendritic cells in vitro and led to NK-cell activation and an increased IFNγ secretion by NK-cells [31]. A study investigating IFNγ as a prognostic marker found that GIST patients who had an increased IFNγ two months after treatment start with imatinib (immunologic responders) had a significantly longer PFS [32]. In vitro, IFNγ upregulates PD-L1 on tumor cells [18]. This could suggest that the combination of imatinib and a checkpoint inhibitor is a promising treatment strategy. 

GISTs have a low CD8+/regulatory T-cell ratio, leading to immunosuppression [10] and increasing the chance of tumor cell survival. In mouse models, imatinib reduced the expression of the immunosuppressive enzyme indoleamine 2,3-dioxygenase (IDO), leading to increased CD8+ T-cells in the tumor [33] and thereby an increased chance of CD8+ T-cell attack of tumor cells. Furthermore, imatinib induced apoptosis in the regulatory T-cells in the tumor, leading to an increased CD8+/regulatory T-cell ratio [33], which is desirable in the oncological treatment of GIST.

One of the strengths of this study is the magnitude of the cohort with GIST patients included independent of disease- and treatment status, leading to a unique patient material. Furthermore, this is a national study. In Denmark, the treatment of GIST patients is standardized across the country. By having a national cohort, selection bias is minimized, which improves generalizability. Additionally, all samples were handled and stored according to national guidelines in Danish CancerBiobank. All samples were analyzed in one batch, excluding variation over time. Despite the strengths, we do face some limitations. The postoperative samples were taken 24 h after surgery. Therefore, one could question if this could represent a patient without evidence of disease. Furthermore, the trauma of undergoing surgery could affect concentration. Moreover, patients were included at different time points during the treatment course, which makes data interpretation challenging.

## 5. Conclusions

In this national prospective study, the prognostic value of plasma PD-1 and PD-L1 concentrations were investigated in patients with GIST independent of disease and treatment status. We found that patients with active GIST have significantly higher plasma PD-1 and PD-L1 concentrations than patients without evidence of disease. Patients with active GIST with a plasma concentration of PD-L1 above the cutoff had a significantly poorer prognosis than those with plasma PD-L1 concentrations below the cutoff. Plasma PD-L1 shows potential as a prognostic biomarker in patients with GIST and should be further evaluated. 

## Figures and Tables

**Figure 1 cancers-14-05753-f001:**
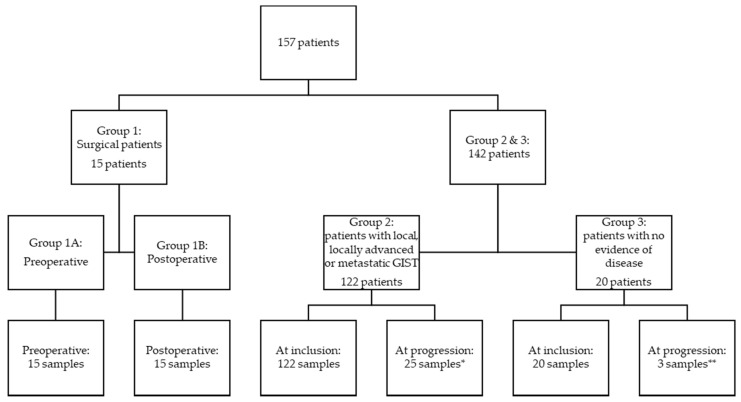
Flow chart of included patients. Group 1: patients with local disease who underwent radical surgery; group 2: patients with local, locally advanced, microscopic, or macroscopic metastatic disease, and group 3: patients without measurable disease (patients radically resected for localized GIST and in adjuvant treatment or patients in surveillance after completed adjuvant treatment). * Thirty-four patients progressed in total, but only 25 progression samples were available from 21 patients. ** Three patients progressed, and all three had a progression sample available.

**Figure 2 cancers-14-05753-f002:**
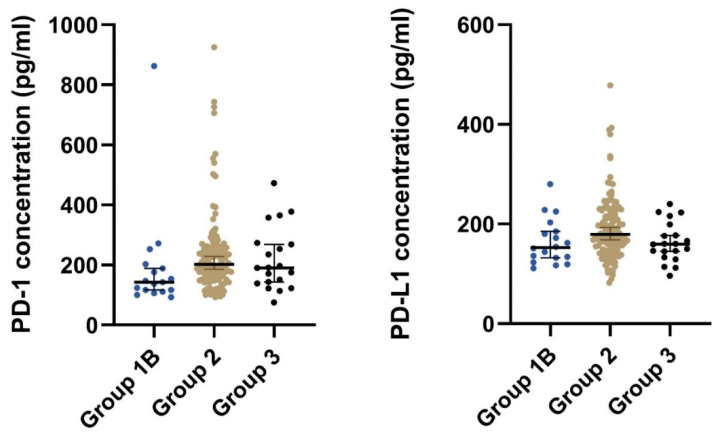
The plot visualizes the plasma concentration of PD-1 and PD-L1 in the group of postoperative samples from patients that had undergone radical surgery (group 1B) and the group of patients with local, locally advanced, or metastatic GIST (group 2). The horizontal lines mark the median values in the two groups. Abbreviations: PD-1: Programmed death protein-1; PD-L1: Programmed Death-Ligand 1.

**Figure 3 cancers-14-05753-f003:**
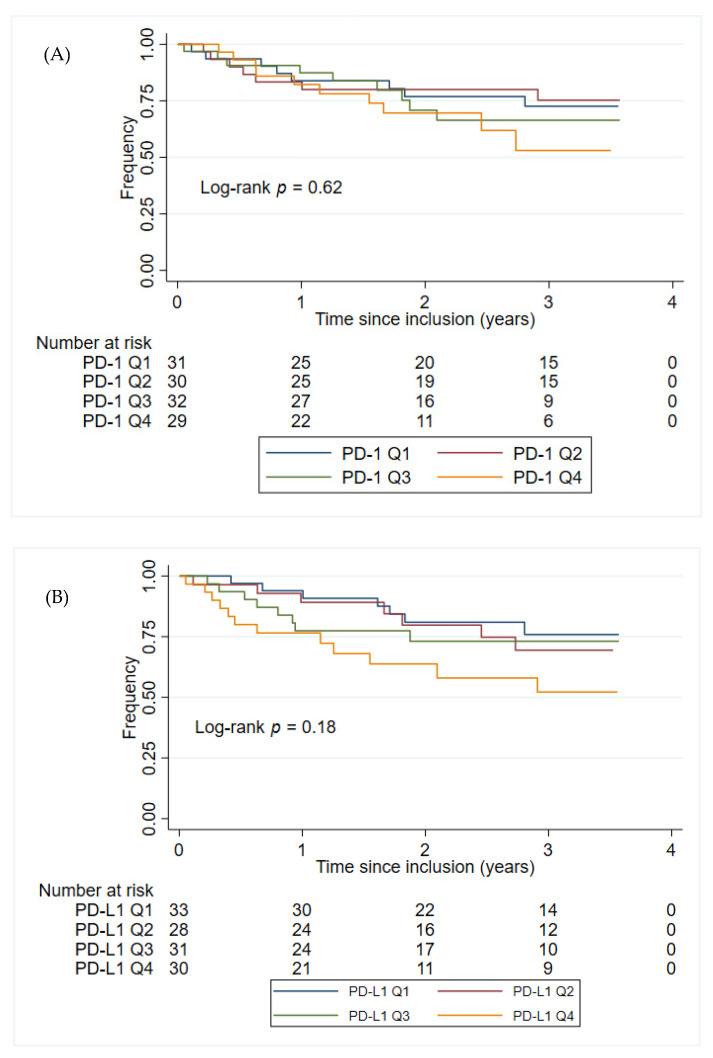
Kaplan–Meier plots for PFS from the time for inclusion of patients in group 2. (**A**) The patients in group 2 were stratified into four groups (Q1–Q4) based on the PD-1 concentration in group 2. PD-1 Q1: ≤ 162 pg/mL; PD-1 Q2: > 162 pg/mL, ≤202 pg/mL; PD-1 Q3: > 202 pg/mL, ≤258 pg/mL; PD-1 Q4: > 258 pg/mL. (**B**) The patients in group 2 were stratified into four groups (Q1–Q4) based on the PD-L1 concentration PD-L1 in group 2. PD-L1 Q1: 149 pg/mL; PD-L1 Q2: > 149 pg/mL, ≤179 pg/mL; PD-L1 Q3: > 179 pg/mL, ≤225 pg/mL; PD-L1 Q4: > 225 pg/mL. Abbreviations: PD-1: Programmed death protein-1; PD-L1: Programmed Death-Ligand 1.

**Figure 4 cancers-14-05753-f004:**
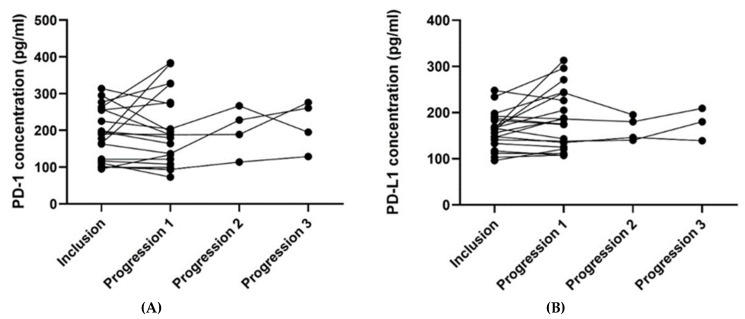
The plots visualize the PD-1 (**A**) and PD-L1 (**B**) concentrations at the time of inclusion and at the time of progression for the 21 patients having an available progression sample. In Figure 4A, one patient’s samples were removed since they were outliers (plasma concentration of PD-1 at the inclusion of 726 pg/mL and 643 pg/mL at progression). In Figure 4B, one patient’s samples were removed since the progression sample was an outlier (plasma concentration of PD-L1 at the inclusion of 215 pg/mL and 751 pg/mL at progression). The outliers in Figure 4A,B were removed since they made it difficult to visualize the other patients’ results. Abbreviations: PD-1: Programmed death protein-1; PD-L1: Programmed Death-Ligand 1.

**Figure 5 cancers-14-05753-f005:**
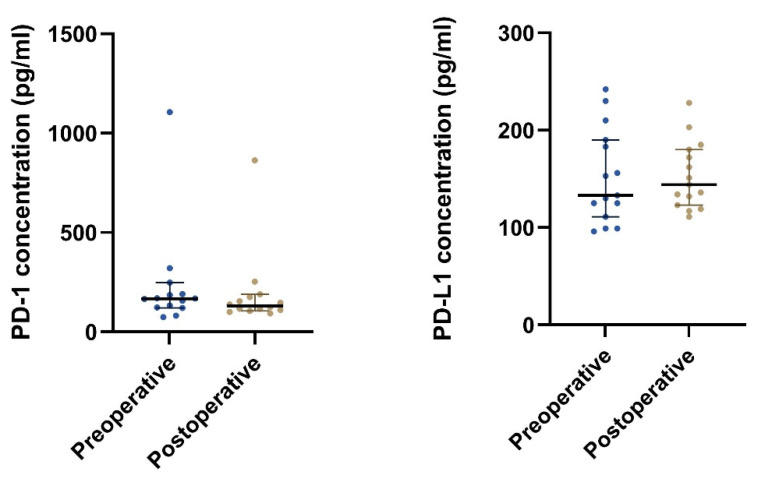
The plot visualizes the plasma concentrations of PD-1 and PD-L1 in the pre- (group 1A) and postoperative (group 1B) samples from patients undergoing radical surgery (*n* = 15). The horizontal lines mark the median values in the two groups.

**Figure 6 cancers-14-05753-f006:**
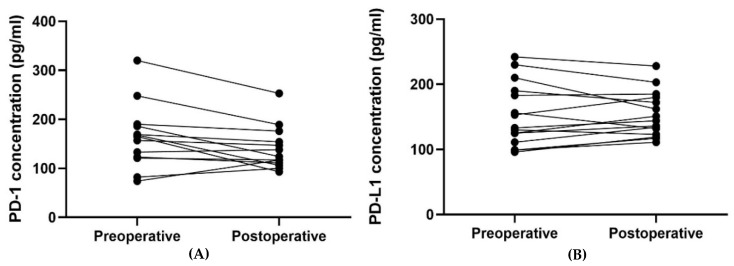
The plot visualizes the plasma concentrations of PD-1 (**A**) and PD-L1 (**B**) in the pre- (group 1A) and postoperative (group 1B) samples from patients undergoing radical surgery (*n* = 15). In Figure 6A, one patient’s samples were removed since they were outliers (plasma concentration of PD-1 preoperative of 1105 pg/mL and postoperative 863 pg/mL) and made it difficult to visualize the other patients’ results.

**Table 1 cancers-14-05753-t001:** Patient and disease characteristics at the time of inclusion for patients undergoing radical surgery (group 1), patients with GIST (group 2), and patients without evidence of disease (group 3).

Patient Characteristics	Patients Undergoing Radical Surgery (Group 1) ^1^, *n* = 15 N, (%)	Patients with Active GIST (Group 2) ^2^, *n* = 122 N, (%)	Patients without Evidence of Disease (Group 3) ^3^, *n* = 20 N, (%)	*p*-Value
Sex Male Female	6 (40.0) 9 (60.0)	65 (53.3) 57 (46.7)	7 (35.0) 13 (65.0)	0.23
Age in years Median (min-max)	73 (44–92)	69 (20–87)	66 (32–81)	0.22
Disease status at inclusion No evidence of disease Local disease Locally advanced disease Microscopic disease ^4^ Metastatic disease	0 (0.0) 14 (93.3) 1 (6.7) 0 (0.0) 0 (0.0)	0 (0.0) 10 (8.2) 28 (23.0) 28 (23.0) 56 (45.9)	20 (100.0) 0 (0.0) 0 (0.0) 0 (0.0) 0 (0.0)	
Treatment at inclusion No treatment Adjuvant Neoadjuvant Lifelong	15 (100.0) 0 (0.0) 0 (0.0) 0 (0.0)	≤3 (≤2.5) 0 (0.0) 18 (14.8) 102 (83.6)	8 (40.0) 12 (60.0) 0 (0.0) 0 (0.0)	

^1^ Patients planned for radical surgery of GIST. ^2^ Patients with local, locally advanced, or metastatic disease. ^3^ Patients radically resected for localized GIST and in adjuvant treatment or patients in surveillance after completed adjuvant treatment. ^4^ Patients with microscopic disease after R1 surgery or metastases surgery etc.

**Table 2 cancers-14-05753-t002:** Plasma PD-1 and PD-L1 concentrations stratified on the patients’ disease statuses.

Disease Status	PD-1, pg/mL		PD-L1, pg/mL	
	Median (P5–P95)	*p*-Value	Median (P5–P95)	*p*-Value
Group 2	202.0 (110.0–540.0)	0.0023 *	179.0 (107.0–294.0)	0.012 *
No evidence of disease ^1^, *n* = 15	131.0 (92.5–863.0)	0.033	144.0 (111.0–228.0)	0.098
Local disease ^2^, *n* = 10	207.0 (93.2–396.0)		190.0 (120.0–380.0)	
Locally advanced disease ^2^, *n* = 28	208.5 (107.0–555.0)		173.5 (98.4–249.0)	
Microscopic disease ^2^, *n* = 28	208.5 (142.0–726.0)		177.0 (102.0–265.0)	
Metastatic disease ^2^, *n* = 56	189.5 (102.0–540.0)		183.5 (115.0–332.0)	

^1^ Postoperative samples from patients radically resected of GIST (group 1B), ^2^ Patients in group 2 * This group is compared with the postoperative samples from patients radically resected of GIST (group 1B) Abbreviations: PD-1: Programmed death protein-1; PD-L1: Programmed Death-Ligand 1; P5: the 5th percentile; P95: the 95th percentile.

**Table 3 cancers-14-05753-t003:** Univariate and multivariate analysis of the plasma PD-1 and PD-L1 concentrations * at the time of inclusion for patients with GIST (group 2).

	Univariate Analysis			Multivariate Analysis **		
	HR	95% CI	*p*-Value	HR	95% CI	*p*-Value
PD-1	1.48	0.75–2.93	0.26	1.55	0.78–3.10	0.21
PD-L1	1.70	0.86–3.38	0.13	1.80	0.90–3.60	0.095

* Group 2 was divided into low or low concentrations based on the median values of PD-1 or PD-L1 in group 2. ** The multivariate analysis included sex and age at inclusion. Abbreviations: HR: Hazard ratio; CI: Confidence interval; PD-1: Programmed death protein-1; PD-L1: Programmed Death-Ligand 1.

**Table 4 cancers-14-05753-t004:** Plasma PD-1 and PD-L1 concentrations at the time of inclusion compared to the first time of progression for patients with an available blood sample at the time of progression.

	Blood Sample at the Time of Inclusion (*n* = 21)	Blood Sample at the Time of Progression (*n* = 21)	
	Median (P5–P95)	Median (P5–P95)	*p*-Value
PD-1, pg/mL	190.0 (95.0–314.0)	188.0 (93.4–384.0)	0.56
PD-L1, pg/mL	163.0 (103.0–234.0)	186.0 (108.0–313.0)	0.062

Abbreviations: P5: the 5th percentile; P95: the 95th percentile; PD-1: Programmed death protein-1; PD-L1: Programmed Death-Ligand 1.

**Table 5 cancers-14-05753-t005:** Plasma PD-1 and PD-L1 concentrations pre- (group 1A) vs postoperative (group 1B) in patients undergoing radical resection for GIST.

	Preoperative (Group 1A), *n* = 15	Postoperative (Group 1B), *n* = 15	
	Median (P5–P95)	Median (P5–P95)	*p*-Value
PD-1, pg/mL	166.5 (73.5–1105.0)	131.0 (92.5–863.0)	0.024
PD-L1, pg/mL	133.0 (96.4–242.0)	144.0 (111.0–228.0)	0.79

Abbreviations: P5: the 5th percentile; P95: the 95th percentile; PD-1: Programmed death protein-1; PD-L1: Programmed Death-Ligand 1.

**Table 6 cancers-14-05753-t006:** Studies investigating PD-L1 expression in Gastrointestinal Stromal Tumor (GIST) specimens.

Study Characteristics	Study			
	By Bertucci F. et al. [17]	By Blakely A.M. et al. [19]	By Zhao R. et al. [18]	By Sun X. et al. [20]
**Year**	2015	2018	2019	2021
**Study type**	Retrospective study	Retrospective study	Retrospective study	Retrospective study
**Patients**	Tumor specimens from 159 patients resected for localized GIST without receiving adjuvant imatinib	Tumor specimens from 127 patients’ GISTs	Tumor specimens from 238 patients undergoing resection of GIST	Tumor specimens from 507 patients undergoing radical surgery
**Type of** **Material used for the PD-L1 analysis**	Tissue	Tissue microarrays	Tissue	Tissue microarrays
**Method for** **PD-L1 analysis**	Whole-genome DNA microarrays (Affymetrix U133 Plus 2.0 and Agilent 44K)	IHC	Real-time RT-PCR	IHC
**GIST PD-L1** **expression rate**	-	69%	-	46%
**PD-L1** **expression was associated with a poor** **prognosis**	No	Yes	Yes	No
**PD-L1** **expression associated with**	A high mRNA PD-L1 expression was associated with low-risk GIST according to AFIP criteria [6]. A high mRNA PD-L1 expression was also related to patients without metastatic relapse.	A high PD-L1 expression was associated with a higher mitotic count and increasing tumor size.	A high PD-L1 expression was associated with a higher relapse rate of GIST. A significantly lower PD-L1 expression was found in patients with very low-, low-, or intermediate-risk GIST compared to high-risk according to NIH consensus criteria [23].	A high PD-L1 expression was associated with a lower mitotic count and smaller tumor size.
**Relation** **between CD8+ T-cells and** **PD-L1** **expression**	Patients with a high PD-L1 expression had a significantly higher CD8+ T-cell metagenes.	The percentage of CD8+ T-cells was inversely related to the PD-L1 expression.	The percentage of CD8+ T-cells was inversely related to the PD-L1 expression.	PD-L1 expression was associated with a high number of CD8+ T-cells

Abbreviations: PD-L1: Programmed Death Ligand-1; IHC: Immunohistochemistry; RT-PCR: Reverse Transcription-Polymerase Chain Reaction; AFIP: the Armed Forces Institute of Pathology criteria [6]; NIH: National Institute of Health consensus criteria [23].

**Table 7 cancers-14-05753-t007:** Studies investigating treatment with checkpoint inhibitors in patients with GIST.

Study Characteristics	Study		
	By Toulmonde M. et al. [27]	By D’Angelo SP et al. [25] and an Expansion Cohort by Chen J.L. et al. [29]	By Singh A.S. et al. [26]
**Year**	2017	2018	2022
**Study type**	Open-label phase II study	Randomized phase II study	Open-label, randomized, phase II study
**Patients** **investigated**	Patients with sarcoma, including GIST (*n* = 10)	Patients with sarcoma, including GIST (*n* = 18)	Patients with advanced or metastatic GIST previously progressed on imatinib (*n* = 36)
**Treatment** **investigated**	Pembrolizumab + cyclophosphamide	Nivolumab vs. Nivolumab + Ipilimumab	Nivolumab vs. Nivolumab + Ipilimumab
**Primary** **endpoint**	The 6-month non-progression rate	The 6-month response rate	The objective response rate > 15%
**Results**	Limited activity in patients with GIST with a 6-month non-progression of 11.1%	The median PFS was 1.5 (Nivolumab) and 2.9 months (Nivolumab + Ipilimumab). Patients with GIST responded poorly to Nivolumab (1 of 9 patients). The 6-month response rate was 0% in both treatment arms.	The median PFS was 2.7 months (Nivolumab) and 1.9 months (Nivolumab + Ipilimumab). The primary endpoint was not met in either treatment group.

## Data Availability

The datasets generated and analyzed in this study are not publicly available. This is in accordance with the rules concerning processing personal data described in the EU General Data Protection Regulation (GDPR) and the Danish Data Protection Act. However, should a researcher be interested in our data, they are welcome to contact us and collaborate. The data can be requested from the corresponding author, charlotte.margareta.brinch@regionh.dk, Herlev & Gentofte Hospital, Borgmester Ib Juuls Vej 1, 2730 Herlev, Denmark.

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
