# Peer review of "The Prognostic Value of Plasma Programmed Death Protein-1 (PD-1) and Programmed Death-Ligand 1 (PD-L1) in Patients with Gastrointestinal Stromal Tumor"

_cancers, 2022, doi:10.3390/cancers14235753_

Round 1
Reviewer 1 Report
Charlotte and co-authors examined the plasma PD-1 and PD-L1 concentrations in patients with Gastrointestinal Stromal Tumour. The results show that Patients with active GIST had significantly higher plasma concentrations of PD-1 and PD-L1 than patients without evidence of disease. Patients with active GIST had the highest plasma concentration of PD-L1 and a significantly poorer prognosis than patients with low concentrations of plasma PD-L1. This investigation is very important for the early prognosis of the patient with that tumor. However, it needs some minor issue need to be addressed.
1. The author should briefly show the assays for measuring the PD-1 and PD-L1 in the section of 2.3 although the author mentioned according to the descriptions from the supplier.
2. The author should show the result and analysis of co-relation between gender and PD-1/PD-L1 concentration although the author show there is no significantly difference. “The sex was not significantly associated with plasma PD-1 concentration (p = 0.056) 178 but significantly associated with plasma PD-L1 (p = 0.0068) concentration, with males hav- 179 ing higher plasma concentrations.”
3, The plot visualizes the plasma concentrations of PD-1 and PD-L1 in the group 1 and 2. The author should show the values of PD-1 and PD-L1 in the group 3.
Reviewer 2 Report
This study presents the characteristics and clinical significance of plasma PD-L1 and PD-1 in patients with GIST and is sound scientific research.
Comment 1: In patient group 2, is high PD-L1 (Q4) an independent prognostic factor
when adjusted for age, sex, and disease status at inclusion (local, locally advanced~~)?
Comment 2: Minor comments. It is hard to understand Table 1 because the lines do not match. Also, in line 107, the sentence should start with a capital letter.
